# Structural multi-colour invisible inks with submicron 4D printing of shape memory polymers

Wang Zhang [1], Hao Wang [1 ✉], Hongtao Wang [1], John You En Chan[1], Hailong Liu[1,2], Biao Zhang[3], Yuan-Fang Zhang[4], Komal Agarwal[1], Xiaolong Yang[5], Anupama Sargur Ranganath[1], Hong Yee Low[1], Qi Ge [6] & Joel K. W. Yang [1,2 ✉]

Four-dimensional (4D) printing of shape memory polymer (SMP) imparts time responsive properties to 3D structures. Here, we explore 4D printing of a SMP in the submicron length scale, extending its applications to nanophononics. We report a new SMP photoresist based on Vero Clear achieving print features at a resolution of ~300 nm half pitch using two-photon polymerization lithography (TPL). Prints consisting of grids with size-tunable multi-colours enabled the study of shape memory effects to achieve large visual shifts through nanoscale structure deformation. As the nanostructures are flattened, the colours and printed information become invisible. Remarkably, the shape memory effect recovers the original surface morphology of the nanostructures along with its structural colour within seconds of heating above its glass transition temperature. The high-resolution printing and excellent reversibility in both microtopography and optical properties promises a platform for temperature-sensitive labels, information hiding for anti-counterfeiting, and tunable photonic devices.

[1] Engineering Product Development, Singapore University of Technology and Design, Singapore 487372, Singapore. [2] Nanofabrication Department, Institute of Materials Research and Engineering, Singapore 138634, Singapore. [3] Frontiers Science Center for Flexible Electronics, Xi'an Institute of Flexible Electronics (IFE) and Xi'an Institute of Biomedical Materials & Engineering (IBME), Northwestern Polytechnical University, 127 West Youyi Road, 710072 Xi'an, China. [4] Digital Manufacturing and Design Centre, Singapore University of Technology and Design, Singapore 487372, Singapore. [5] National Key Laboratory of Science and Technology on Helicopter Transmission, Nanjing University of Aeronautics and Astronautics, 210016 Nanjing, China. [6] Department of Mechanical and Energy Engineering, Southern University of Science and Technology, 518055 Shenzhen, China. ✉email: whchn@live.cn; joel_yang@sutd.edu.sg

**4D** printing[1–3] brings together the design flexibility of 3D printing with stimuli responsive properties of its constituent materials. It continues to generate excitement in diverse fields, e.g., soft robotics[4–6], drug delivery[7,8], flexible electronics[9,10] and tissue engineering[11]. Commonly used printing methods for 4D printing include direct ink writing[3,12], Polyjet[13,14], Digital Light Processing (DLP) lithography[15,16] and Stereolithography (SLA)[17,18]. The material and lithographic challenges inherent to these methods limit the minimum feature size of printed structures to ~10 μm[19]. At an order of magnitude smaller, submicron scale features that interact strongly with light have yet to be systematically explored in 4D printing.

The motivation to improve print resolution is fuelled by applications in optics, e.g. structural colour generation[20,21], temperature-sensitive passive labels and colorimetric pressure sensors, all of which require submicron resolution and precision. Traditionally, different structures such as gratings[22,23], thin films and multilayers[24,25], localised resonance structures[26,27] generate fixed colours without the use of pigments. Recently, dynamically reconfigurable colours have gained interest, where optical responses of nanostructures can be tuned either by changing the refractive index[28–30] or dimensions[31,32] of the structures. Of these methods, tuning the dimensions of the optical devices by shape memory polymers (SMPs) is of interest due to their relatively short response times (seconds to minutes depending on actuation temperature[33]). Distinct from the pattering of SMPs through nanoimprinting[34–36] and self-assembly[37–39], our use of 3D printing introduced here will lead to direct patterning of complex structures at will, bringing together fields of mechanical and optical metamaterials with local control of properties, e.g., colours[40], phase and Young's modulus.

To print finer 3D structures, we develop a new resist suited for two-photon polymerization lithography (TPL)[6,41,42]. Here, photo-initiators in a liquid resin are excited by a two-photon absorption process within the focal region of a femtosecond laser. Polymerisation and crosslinking then ensue. Printed features as small as ~10 nm can be achieved under specific conditions[43]. Due to the high resolution it provides, TPL has been used to print different stimuli responsive materials such as hydrogels[6,44], liquid crystal elastomers[41,45], magnetic nanoparticles embedded resists[46,47], silicon functionalized monomers[48] and other examples printing[49]. Recently, hydrogel photoresists have been used for tunable photonic devices. Marc et al.[50] used a cholesteric liquid crystals (LC)-based hydrogel resist to change colours at microscale. In their work, colours were tuned within a limited range by changing the intrinsic periodicity of chiral LC. Tao et al.[51] demonstrated a hydrogel based reconfigurable photonic crystals exhibiting colour shifts in the presence of humidity. In contrast to previous works reporting colour shifts, our work investigates submicron 4D printing where large and rapid visual responses are achieved as nanostructures recover from a flattened (colourless) state to an upright (colourful) state. The visual effect is analogous to a letter written in multiple colours of invisible ink where secret information is revealed, e.g., with the application of heat. We tackled two key challenges in (1) developing and characterising new stimuli responsive resists suitable for TPL, and (2) designing and fabricating robust 3D photonic structures capable of rapid recovery after being flattened.

In this work, we meet these challenges by additive manufacturing of SMP for programmable colour generation. We developed and characterised a SMP photoresist suited for TPL based on Vero Clear[13], which is an optically transparent thermosetting polymer resin containing acrylate functional group. We performed resolution tests achieving ~300 nm half pitch gratings and measured the thermodynamic properties of the new resist to determine an optimal composition for robust mechanical performance. A range of structural colours were achieved by controlling the geometry of the crosslinked SMP structures at the submicron level. We realised colour switching behaviour by heating and deforming (i.e., programming) the printed structure at a temperature higher than the glass transition temperature ($T_g$). Remarkably, the deformed nanostructures exhibited excellent recovery when heated above its composition-adjustable $T_g$. This reversibility in both structural features and optical responses demonstrate significant promise for submicron scale additive manufacturing of SMPs.

## Results

**Two-photon polymerization lithography of shape memory polymer.** We designed structures consisting of a base layer with submicron-scale grids on top of it, as shown in Fig. 1. Due to the interaction of these nanostructures with light, i.e., scattering and interference, the 3D printed structures function as colour filters, preferentially transmitting certain wavelength ranges of an incident white light illumination. Colours depend sensitively on the geometric parameters of the grid, i.e., grid height $h_2$, and grid linewidth $w_1$, while it is less sensitive to pitch $w_2$ and the thickness of the base layer $h_1$. By printing in SMP, we realise a 4D effect, with the ability to change its geometry and optical properties in response to temperature variation as a function of time.

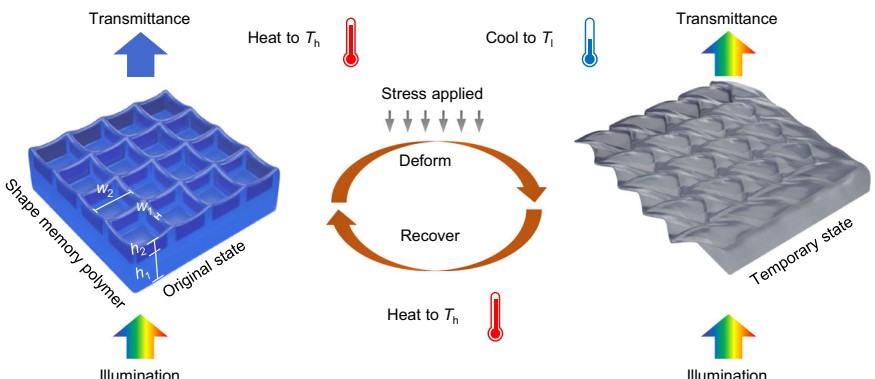

**Fig. 1 Schematic of colour and shape change of a constituent nanostructured element of the "invisible ink" 3D printed in shape memory polymer (SMP).** The as-printed structures with upright grids (left) function as a structural colour filter that transmits only a limited wavelength range of visible light. Deformation of the structures at elevated temperature flattens the nanostructures (right) rendering it colourless, where it remains in an invisible state after cooling to room temperature. Heating recovers both the original geometry and colour of nanostructures, leading to a submicron demonstration of 4D printing.

To achieve the shape memory effect, the print is first deformed at a temperature $T_h$ higher than the SMP's glass transition temperature. While keeping the external load, the temperature is decreased to $T_1$ ($<T_g$) as the print transitions from a soft rubbery state to a stiff glassy state. Here, in the altered flattened geometry the print loses its colour, rendering the print "invisible". The temporary configuration is achieved at $T_1$ after releasing the external load as the polymer chains are "frozen" at its glass state. The print finally recovers to its original geometry and colour when heated back to $T_h$ where the polymer chains regain its entropic elasticity.

A Vero Clear[14] based SMP photoresist was developed (see Methods section and Supplementary Fig. 1 for the preparation process). To test the resist, we placed a droplet of it onto a fused silica glass substrate and exposed using in the commercial TPL system Nanoscribe GmbH Photonic Professional GT using the "dip-in" configuration[40,52,53]. During exposure, photo-initiators at the focal point of the objective are activated by two-photon excitation from the femtosecond pulsed laser at 780 nm wavelength, leading to polymerisation of the resist into solid structures. After exposure, the uncured resist was removed using a development process (see Methods section for details). The characterisation of the photoresist is provided in Supplementary Information part 2. With the characterised resist, we were able to print samples with linewidth of ~280 nm (Supplementary Fig. 2e) and conservative minimum resolvable pitch of 600 nm (i.e. 300 nm half pitch, Supplementary Fig. 2f). This resolution for additive manufacturing of a SMP is an order of magnitude higher than traditional high-resolution printing methods such as DLP[15,16] and SLA[17,18].

**Characterisation of colours**. We next investigate the different colours achieved by the grid structure shown in Fig. 1 and how it depends on the various design parameters. As $h_1$ only affects the phase of light and does not contribute to the change of colour, it is fixed at ~4 μm by fixing the laser power (35 mW), write speed (2 mm/s) and number of writing layers (10 layers) to raise the grids above the substrate making it easier to compress. The two parameters $h_2$ and $w_1$ can be varied by controlling the write speed, laser power and number of grid layers. See Supplementary Information part 3 on a discussion of laser power and write speed to get different colours. Figure 2a shows a transmittance optical micrograph of a colour palette as a function of write speed $w_1$ and nominal height $h_2$ for a range of laser power (30–35 mW) and fixed pitch $w_2$ (see Table S1 for the fabrication parameters). The corresponding transmittance spectra for Fig. 2aI were measured using a CRAIC microspectrophotometer and mapped onto the CIE 1931 chromaticity diagram in Supplementary Fig. 4, demonstrating a reasonably wide range of colours. To study the effect of pitch $w_2$ on colour, we fabricated structures of constant nominal height $h_2$ of 1.8 μm and varied its pitch. The transmittance spectra for structures with different pitches are shown in Supplementary Fig. 5a (see the corresponding SEM images in Supplementary Fig. 5b–h). When $w_2$ is 1 μm, the adjacent lines are fused together during the polymerisation process, and produce a nearly transparent patch. When $w_2$ is 3 μm, the gaps are too wide, leading to low colour saturation. Thus, a gap of 2 μm was chosen for the remainder of our studies. Here the minimum resolvable pitch $w_2$ is larger than that in Supplementary Fig. 2f, which could be caused by the proximity effect while printing multilayer girds. It should be noted that the experimental obtained colour gamut in Fig. 2a can be further extended by decreasing the pitch $w_2$ to increase the colour saturation.

Figure 2b shows representative SEM images for grids with different nominal heights at a fixed write speed of 1 mm/s and laser power of 30 mW (black rectangle in Fig. 2a). Generally, the central region of the grid remains stable with increasing height of the grid, while the corners start to collapse due to resist shrinkage and lack of support structures. The collapsed structures account for nonuniformity in colour at the edges of the square patches in Fig. 2a. As both the laser power and write speed were kept constant, the linewidth of the grid does not change with increasing nominal height as shown in the top view SEM images in Fig. 2b. Both the linewidth $w_1$ and height $h_2$ for the grid structure in the black box are plotted in Fig. 2c. The height increases at an interval of 100 nm per layer, which is less than the nominal layer height (300 nm as shown in Table S1) as a result of shrinkage during the writing and development (rinsing) process.

To study the influence of write speed on the structure, Fig. 2d shows representative SEMs for structures with different write speeds and constant nominal height inside the red box in Fig. 2a and the measured dimensions are summarised in Fig. 2e. The linewidth decreases from 630 nm to 285 nm as the increase of write speed from 0.5 mm/s to 1.1 mm/s, while the height of the grid decreases from 1990 nm to 980 nm. Supplementary Fig. 6 shows the SEM images and measured dimensions for structures with different laser power in Fig. 2a. The effect of write speed and laser power on the structure size can be attributed to the roughly relation curing spot size ~laser power[2]/write speed. With increasing write speed and decreasing laser power, the energy received per unit area decreases, leading to less polymerisation and an effectively smaller curing spot size, resulting in a narrower linewidth and shorter structures.

Though we have previously reported high aspect ratio pillars as colour generating structures[53,54], this is the first time that we are experimenting with grid structures. The grid structures provide greater mechanical stability needed in this study. To gain some understanding of the colour generating nature of this design, we measured the transmittance spectra of the grid structures inside the black box and performed finite difference time domain (FDTD) simulations to obtain the spectra (Fig. 3a). A comparison of the measured and simulated data is provided in Supplementary Fig. 8. The simulation results show a good qualitative agreement with the experiment. The discrepancies could arise from the idealised structures used in the simulations that do not account for rounded edges. A redshift effect of the spectrum position is observed in Fig. 3a and shown by the dotted arrows. The simulated near-field electric field phase and amplitude for one structure (fabricated with the laser power of 30 mW, write speed of 1 mm and nominal grid height of 2.7 μm) at wavelength of 490 nm and 710 nm (corresponding to the marked dip and peak of the spectrum in Fig. 3a) are shown in Fig. 3b, c, respectively. The incident planewave passes through the base layer without scattering. After propagating through the submicron grids, the wave front is delayed compared to the wave fronts that pass through the air gap. As seen in the near-field phase plot in Fig. 3b, the regions within and directly above the grid lines appear to accumulate phase faster than the region in between. The interference of these two regions of transmitted light causes some focusing and redistribution of the energy flow of light (see corresponding electric field amplitude of the near field in Fig. 3c). For the peak and dip positions, the constructive interferences occur at different parts above the structure. The far-field energy distribution can be obtained by performing the near-field-to-far-field transform. Figure 3d, e present the normalised far-field electric field amplitude within the objective collection angle (CA) for the dip and peak position respectively, corresponding to experimental observation with objective lens adopted in our experiment (the NA = 0.2, CA = 11.5°). The integration of the far field electric intensity within the collection angle results in the transmittance spectra dip and peak shown in Fig. 3a.

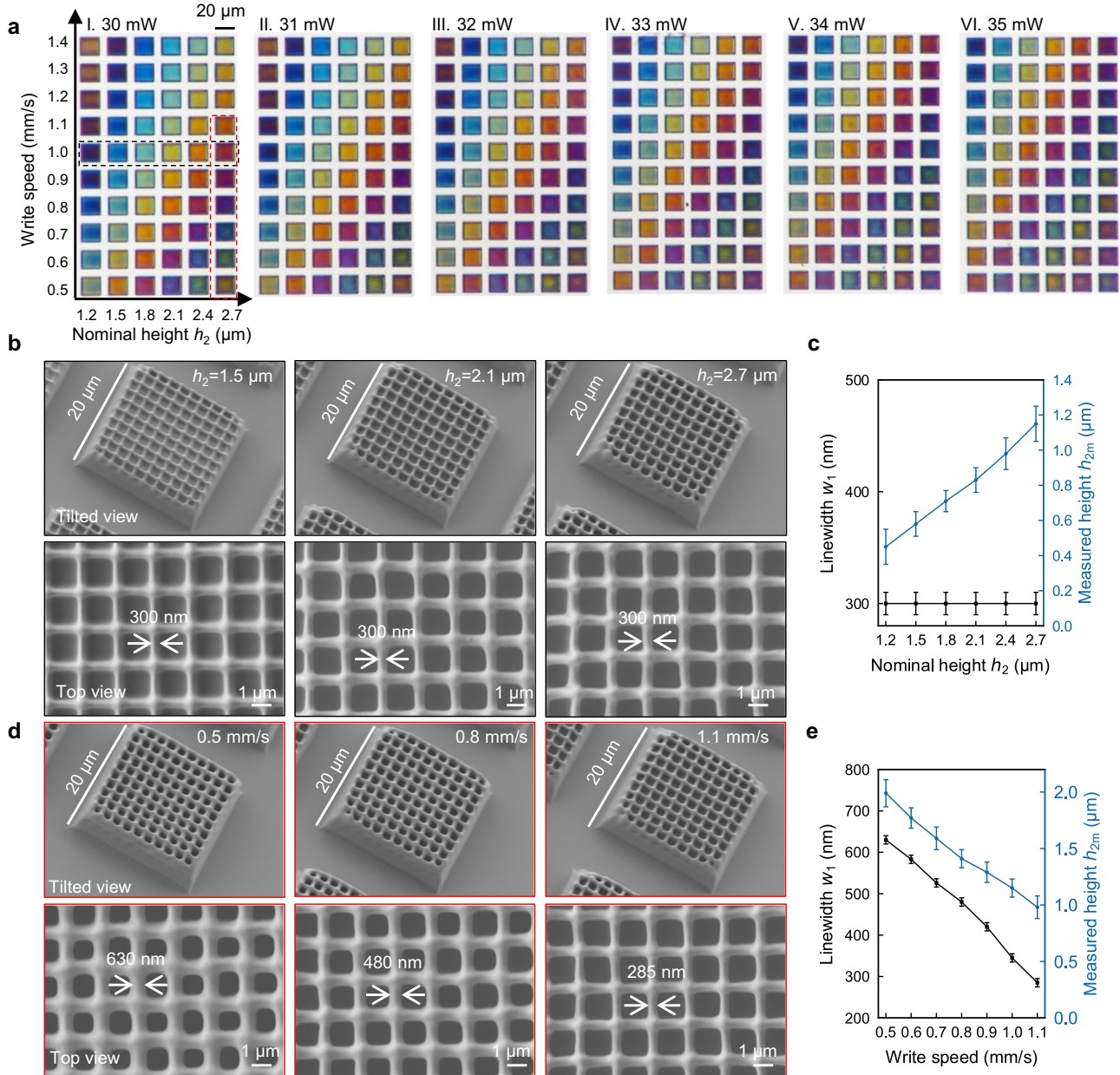

**Fig. 2 Optical and scanning electron micrographs (SEM) of as-printed structures. a** Optical transmittance micrographs of a printed colour palette for a constant pitch of 2 μm but varying write speed and nominal height $h_2$ for a range of laser power (I–VI laser power 30–35 mW respectively). **b** SEM images of grid structures with different nominal height $h_2$ for constant write speed of 1 mm/s and laser power of 30 mW. **c** Measured grid linewidth $w_1$ and height $h_{2m}$ as a function of nominal height; **d** SEM images for grid structure with different write speeds for fixed nominal height of 2.7 μm and laser power of 30 mW. **e** Measured grid linewidth and height as a function of write speed. Values in **c** and **e** represent mean and the error bars represent the standard deviation of the measured values ($n = 5$).

As discussed in Fig. 2, write speed and laser power affect both linewidth $w_1$ and grid height $h_2$. To study the influence of $w_1$ alone on colour, we simulated the spectra for a fixed height $h_2$ of 0.9 μm and varied the linewidth $w_1$ (Fig. 3f). As the linewidth increases from 200 nm to 500 nm, the transmittance dip redshifts from 450 nm to 600 nm, resulting a shift of the transmitted colour from yellow to blue. This effect could be risen from the increase of effective refractive index as the increase of $w_1$. See Supplementary Information part 10 for a detailed discussion about this relation. The redshift effect of the measured spectra as the increasing of laser power and decreasing of write speed in Supplementary Fig. 9c could be explained by the increase of effective refractive index. This result suggests that one could also achieve large colour variation simply by varying the width of the structures.

**Submicron scale shape memory effect.** To study the shape memory effect of the submicron scale structure, we printed a colour palette (Fig. 4aI), with a fixed nominal height of 1.8 μm, but varied the laser power and write speed from 30 mW to 35 mW horizontally with a step of 0.5 mW and 0.6 mm/s to 1.1 mm/s vertically with a step of 0.05 mm/s respectively. Doing so also generates a broad range of colours due to variation in both width and height of the structures. The structures were then heated above its $T_g$ to 80 °C using a heat gun. Under the high

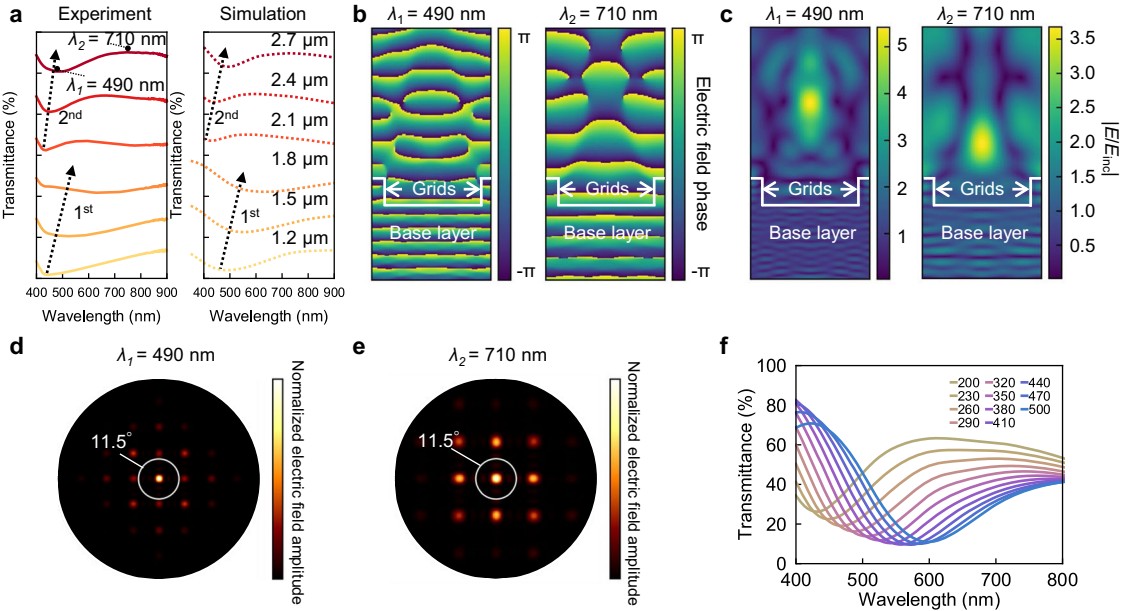

**Fig. 3 Finite difference time domain (FDTD) analysis of the grid structure. a** Measured and FDTD simulated transmittance spectra of structures with different nominal height $h_2$ (from the black dashed rectangle in Fig. 2a ranging from 1.2 μm to 2.7 μm. 1st and 2nd represent the first and second order resonance dip respectively). Marked positions $\lambda_1 = 490$ nm and $\lambda_2 = 710$ nm are used for FDTD field analysis in Fig. 3b–e. **b, c** Cross section view of near-field normalised electric field phase and amplitude for a grid structure (laser power: 30 mW, write speed: 1 mm/s, nominal grid height: $h_2 = 2.7$ μm) at dip transmittance 490 nm and peak transmittance 710 nm wavelength respectively ($|E/E_{inc}|$ represents the normalised electric field amplitude). **d, e** Top view of far-field normalised electric field amplitude for the above grid structure at dip transmittance 490 nm and peak transmittance 710 nm wavelength respectively; the white circle represents collection field for the microscope used in this work (NA = 0.2, CA = 11.5°). **f** Simulated transmittance spectra for structures with different linewidth $w_1$ (the colours of the spectrum lines were mapped from the corresponding spectra).

temperature, a stress of ~500 kPa was applied using a metal block on the surface of the structure. Then the sample was cooled down to room temperature in air (in ~30 s) with the metal block maintained. Upon removal of the load, the deformed structures appear transparent as shown in Fig. 4aII. All the colours were recovered when the sample was heated up again to 80 °C by the heat gun (Fig. 4aIII). The recovery process occurs within seconds due to the rapid response of the SMP. The detailed setup of this programming process is provided in Supplementary Fig. 10. We compared the spectra of three different colours (marked as 1–3 in Fig. 4a) before and after the programming and recovery processes, shown in Fig. 4b. The spectra of the original and recovered colours are almost identical except for a small but systematic blueshift, indicating a good recoverability of our 4D printing. The compressed structure exhibited high transmittance ~80% across the visible spectrum, leading to a transparent appearance devoid of colour.

SEM images (Fig. 4c) of structure 2 were taken at the three states (as printed, after compression and after recovery) showing the change in nanostructure geometry and revealing the underlying mechanism of colour variation. The original grids appear as regular squares, with light scattering and colour generating properties, as explained above. However, the compressed structures appear as irregular quadrangles, with the pits in grid completely closed up due to buckling and collapsing of the walls of the grid into a fish-scale pattern. This configuration is expected to result in weak scattering in the visible spectrum, due to the reduced height of the structures and lack of clear structure definition.

As the structures have been squeezed to the point of contact between surfaces, one would have predicted that this deformation was irreversible. Experience with collapsed nanostructures from capillary forces teaches us that the stiction Van der Waals forces

will keep these nanostructures together[55]. Yet, once heated above $T_g$ again, the grids recover to regular squares again due to the shape memory effect. The top view of structure 2 during the programming process is given in Fig. 4c, in which the linewidth before deformation and after recovery matches well, indicating a good shape recovery effect. It should be noted that as there was no support on the edges of grids, the walls along the edge might have been too thin to overcome the stiction forces and could not recover, as shown in the tilted view SEM image in Fig. 4cIII. This irreversible damage account for some of the dark corners and edges of the recovered structures in Fig. 4aIII.

To further understand the programming process, Fig. 4d shows spectra for a structure programmed into different degree of flatness, and the corresponding SEM images are given in Fig. 4e. When the structure is slightly flattened (Fig. 4eII), the grids configuration is similar to the original one (Fig. 4eI), and there are only gentle shifts of both colour and spectrum (Fig. 4d). While the structure is further flattened, the gaps between the grids are filled (Fig. 4eIII–IV), leading to high transmittance of light wavelength in the whole visible range and a transparent appearance (Fig. 4d). The robustness of the programming process is checked in Fig. 4f by programming a structure for four times, the spectra for recovered structures in different cycles match well, indicating a good repeatability of the shape memory effect. To quantify the influence of stress on the programming process and determine the breaking point, we performed an additional programming process (Supplementary Information part 12) using a nanoimprint machine, which enables more control and a wider range of pressures (Supplementary Fig. 11a). As the applied stress increases to 26 psi (~179 kPa), the structure turned colourless (Supplementary Fig. 11b). While a high stress of up to 196 psi (~1351 kPa) leads to the irreversible collapse around the edges (Supplementary Fig. 11c). Note that further in situ micron and

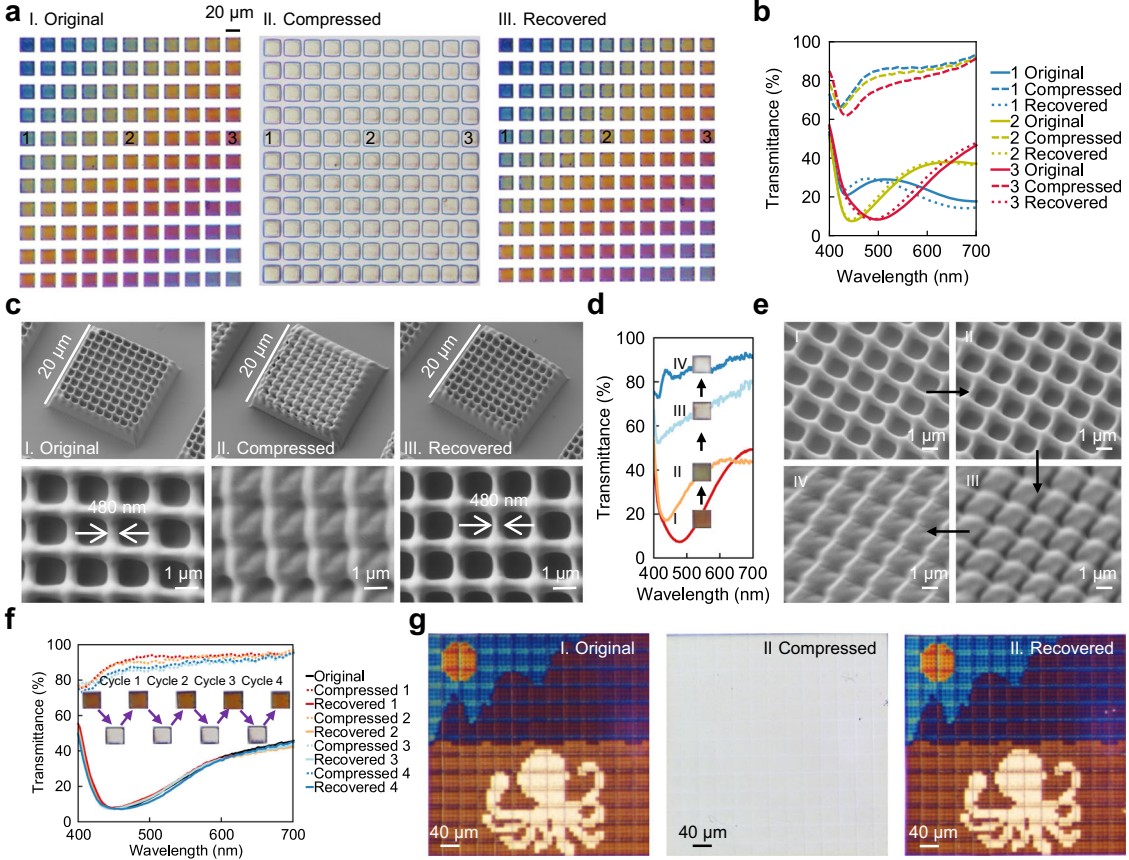

**Fig. 4 Submicron scale shape memory effect. a** Different colours as printed, compressed and recovered respectively, observed by the objective lens (NA = 0.2, CA = 11.5°) at the transmittance mode (in the colour palette, the laser power was varying from 30 to 35 mW with a step of 0.5 mW in the transverse direction, and the write speed was varying from 0.6 to 1.1 mm/s with a step of 0.05 mm/s in the vertical direction). **b** Comparison of measured spectra of three different grid structures (marked as number 1, 2, 3 in Fig. 4a) as printed, after programming and after recovery. **c** Tilted (30° tilt angle) and top view of SEM images before and after programming and after recovery. **d** Measured spectra and **e** SEM images for a structure programmed into different degree of flatness. **f** Measured spectra of a grid structure for four programming cycles. **g** The painting as printed, compressed, and recovered, respectively.

nano scale compression experiments[56] at controlled temperatures will need to be conducted to directly determine the relation between stress and strain during deformation of the structure. Based on this programming process, the repeatability of the shape memory process was further checked, and no obvious change of colour was observed even after 10 programming cycles (Supplementary Fig. 11b). To further increase the number of program-recovery cycles (e.g., beyond 1000 cycles), other specific chemicals such as monomers with strong π–π interactions and massive chain entanglements[57] could be considered in future studies.

To demonstrate the potential application, we printed an image of an art piece by one of the authors, depicting an octopus in foreground with a mountainous landscape in the background (Fig. 4gI). This image comprised 52 × 52 pixels with each pixel designed as 10 μm. The print was then programmed into a transparent featureless image as shown in Fig. 4gII, using the process in Supplementary Information part 11. Here both the octopus and its surrounding have become invisible. Upon heating, the painting recovers to its original state again as shown in Fig. 4gIII. This demonstration could offer a simple way to monitor whether a material has been subjected to high temperatures, which is of importance in the transportation of sensitive goods. Supplementary Fig. 12 presents the SEM images of the top left corner of the original painting. In Supplementary Fig. 12a, different parts were printed by different nominal heights, resulting in different colours in Fig. 4g. To make the whole

structure stable, the lines along two adjacent write fields were written twice, leading to wider lines comparing to lines within one write field as shown in Supplementary Fig. 12b. The grids near the boundaries are stretched to be wider, which results in slightly different colour along the boundaries of different write fields. To overcome this issue, some more stable photoresist with higher stiffness and less shrinkage should be developed in the future.

## Discussion

SMPs based 4D printing is widely studied due to its wide accessibility of materials and simple way of programming. In this work, we used TPL to pattern the characterised SMP photoresist, extending the resolution to 300 nm scale. The resolution of the SMP photoresists are already very high, but it is not yet at the resolution capabilities of the TPL system. Further improvements in resolution could be achieved by tweaking its composition, bringing it to the level of commercially available photoresist such as IP-Dip (~100 nm feature size[58,59]). Due to the shape memory effect at submicron scale, both microtopography and optical properties of the structures can recover well. In some situations where it is not very convenient or even possible to apply load and heat, other methods to induce strain can be considered, e.g., voltage, light and magnetic field to achieve contactless and precise control of the programming. Also, micron

and submicron scale mechanical tests[56,60,61] considering the degree of polymerisation[62], alignment of the polymer chains[63], and size effects[64] should be implemented to understand the micromechanical behaviour of the print. These limitations could be tackled in future work.

In conclusion, we demonstrated the concept of submicron scale 4D printing of shape memory polymer with the application for multi-colour invisible inks by two-photon polymerization lithography of the custom-tailored photoresist. Due to the flexible tunability of the design variables by additive manufacturing, different colours can be easily obtained by varying the printing parameters such as laser power, write speed and nominal height of grids. The printed structures can switch colour stably and rapidly by the programming process. Though there are some key challenges in this area, we believe that this approach opens up potential applications of 4D printing in fields that require high-resolution structures and precision such as optics and sensors.

## Methods

**Preparation of photoresist**. 0.9 g of 2-hydroxy-3-phenoxypropyl acrylate (HPPA) and 0.1 g of Bisphenol A ethoxylate dimethacrylate (BPA) were mixed together and stirred by a vibration generator system for 5 min to get homogeneous solution. 20 mg of diphenyl (2,4,6-trimethylbenzoyl) phosphine oxide (TPO) was added to this solution and stirred by a magnetic rotor for 1 h to get the elastomer resist. The elastomer resist was mixed with Vero Clear at different mass fractions and stirred for 5 mins to get the SMP photoresist. The photoresist was kept in brown glass bottle at 22 °C to avoid polymerisation before printing. Vero Clear was purchased from Stratasys, while HPPA, BPA and TPO were purchased from Sigma-Aldrich. The chemicals were used as received.

**Two-photon polymerization lithography**. Before writing, a fused silica substrate ($25 \times 25 \times 0.7$ mm$^3$, refractive index = 1.46) was cleaned with IPA solution in ultrasound for 2 mins. The substrate was baked at 120 °C for 10 mins on a hotplate, then cooled to room temperature. Then the substrate was spin coated with TI PRIME adhesion promoter (MicroChemicals GmbH, Germany) at 2000 rpm for 20 s. The substrate was again baked at 120 °C for 2 mins on a hotplate, then cooled to room temperature. A drop of photoresist was placed onto the coated side of the substrate, then the substrate was transferred to a two-photon lithography system (Photonic Professional GT, Nanoscribe GmbH, Germany). A ×63 NA1.4 oil immersion objective lens in Dip-in Laser Lithography (DiLL) configuration was used. The laser power and write speed were set to 30–40 mW and 0.5–2 mm/s, respectively. After writing, the substrate was immersed in propylene glycol monomethyl ether acetate (PGMEA) for 5 mins, isopropyl alchohol (IPA) for 2 mins and deionized water for 1 min to remove the uncured resist. The substrate was blown with a N$_2$ gun to remove the water, then cured with UV exposure of 365 nm and ~1 J/cm$^2$ for 10 mins (UVP® CL-1000® Ultraviolet Crosslinkers, USA). Finally, the substrate was kept in a clean container for 2 days to release residual stress.

**Materials characterisation**. The dynamic mechanical analysis tests were conducted on a DMA tester (TA Instruments, Q800 DMA, U.S.) in the tension film mode. After erasing thermal history at 80 °C for 5 min, DMA tests started from 80 to 0 °C at a cooling rate of 3 °C/min. For the test of pure elastomer, the test was started at 40 °C to avoid rupture. The dimensions of the testing samples were 6 mm × 15 mm × 0.5 mm and was prepared by curing the photoresist in a Teflon mould in the UV oven with a power of ~1 J/cm$^2$ for 10 mins.

The uniaxial tensile experiments were conducted on the DMA tester (TA Instruments, Q800 DMA, U.S.). The samples were prepared by the same method as above. The test was conducted using the stress control mode with a stress rate of 2 MPa/min. The temperature was controlled to be 20 °C higher than the glass transition temperature ($T_g$) for each composition.

The viscosity tests were conducted on a Discovery Hybrid Rheometer (DHR2, TA instruments Inc., UK) with an aluminium plate geometry (20 mm in diameter), with frequency ranging from 10 to 4000 Hz. The temperature was precisely controlled to be 22 °C by a Peltier system. The plate gap was set as 100 μm.

The SEM images was taken with a JSM-7600F Schottky Field Emission Scanning Electron Microscope (JEOL, Japan) using a voltage of 5 kV. Before the test, the samples were sputtered with gold in vacuum for 60 seconds with a current of 40 mA at the control gas manual mode.

The height of the structures was measured by a Profilometer KLA Tencor D-600 (KLA Inc., U.S.). The scan speed was 0.01 mm/s and the stylus force was 1 mg.

The refractive index of the photoresist needed for FDTD simulation was measured by an EP4 Ellipsometer (ACCURION, Germany). A wafer substrate was cleaned with IPA and then baked on a hotplate at 130 °C for 10 min. Then the resist was spin coated on the substrate with a speed of 7000 rpm for 1 min. Then

the substate was baked again on the hotplate 130 °C for 3 min. Afterwards, the sample was used for the measurement of the ellipsometry angles Δ and Ψ of the photoresist. The refractive index $n$ and extinction coefficient $k$ were fitted based on the measured Δ and Ψ using the Cauchy dispersion function as shown in Supplementary information part 8.

**Optical measurements**. Transmittance spectra were measured using an objective lens (NA = 0.2, CA = 11.5°) on an optical microscope (Nikon Eclipse LV100ND) with a CRAIC 508 PV microspectrophotometer and a Nikon DS-Ri2 camera. Samples were illuminated with a halogen lamp. The spectra (transmittance mode) are normalised to the transmittance spectrum of fused silica glass, which is measured under the same conditions as for the sample.

**Numerical simulation**. FDTD simulation to calculate the theorical spectra was conducted with a commercial software (FDTD, Lumerical Solutions). The dimension profile for the simulation was obtained from the SEM images and the Profilometer. In FDTD simulation, planewave was injected from bottom of grids and base layer, then electric field and corresponding power components were collected and followed by a near-to-far-field transform process. The final spectra were obtained by integration of the energy within the collecting angle of the objective lens used in measurement.

## Data availability

All data are available from the corresponding author upon reasonable request.

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

## Acknowledgements

J.K.W.Y. acknowledges funding support from the National Research Foundation (NRF) Singapore, under its Competitive Research Programme award NRF-CRP20-2017-0004 and NRF Investigatorship Award NRF-NRFI06-2020-0005. B.Z. acknowledges the National Natural Science Foundation of China (No. 51903210), Natural Science Basic Research Program of Shanxi (Program No. 2020JQ-174), the Fundamental Research Funds for the Central Universities (No. 31020190QD015). H.Y.L. acknowledges funding support from Singapore Ministry of Education T2MOE1720.

## Author contributions

W.Z. conceived the idea, designed the experiments, characterised the photoresists, and fabricated the samples with the assistance from Hao W.; H.T.W. performed the FDTD simulation. J.Y.E.C., H.L., B.Z., Y.F.Z., K.A., X.Y. and A.S.R. assisted the fabrication and characterisation. J.K.W.Y. supervised the research in discussion with H.Y.L. and Q.G. All authors contributed to writing and revision of the manuscript.

## Competing interests

The authors declare no competing interests.
