## [Peer Review File · Nature Communications]

Reviewers' Comments:

Reviewer #1:

Remarks to the Author:

The paper by Zhang and co-workers entitled 'Structural Multi-Colour Invisible Inks with Submicron 4D Printing of Shape Memory 2 Polymers' describes an exciting new development of nanostructured polymer films that consist of sub-micron features that give rise to a reflected colour. This structure-induced colour can be switched off by subjecting the material to mechanical pressure so as to flatten the film, which can be subsequently recovered by heating the sample to the glass transition temperature. Two photon lithography is used to achieve the submicron features when used with a new polymer resist developed by the authors. Previously, the same team has demonstrated structural colour from high-aspect ratio pillars, but the results presented in this manuscript appear to show a sufficiently novel approach using the grid-like structures.

The study appears to be of a very high quality: the manuscript is very well written, free from jargon, and easily understandable. The figures have also been assembled and prepared with great care. A range of structural colours is demonstrated, which is rather impressive, and there is a good level of detail in terms of the variation in the observed colour with the architecture of the grid and the authors have carried out substantial work to interpret the results. The authors have clearly thought about how the structural colour is influenced by the dimensions of the grid. The experimental data of the structural colour for different nanostructure geometries are also supported, qualitatively by FDTD simulations.

I recommend this manuscript for publication, but suggest that the following minor points are considered first:

- 1) To remove the colour by applying pressure, the material has to be first heated and then cooled in the presence of an external load, which appears to be somewhat limiting from an applications point of view. Could the authors comment on this? It could be that the key application is that this offers a neat way to validate whether a material has been subjected to high temperatures, which is of importance in the transportation of sensitive goods.
- 2) The statement on line 114 is rather strong. I am not certain this is the lowest feature size reported, although I cannot immediately point to the literature demonstrating smaller feature sizes. This needs to be checked thoroughly.
- 3) Presumably the polarisation of the writing laser was not of importance in terms of the features that were written?
- 4) Why was the small power range (30-35 mW) chosen?
- 5) I found the text in Figures 3b and 3c rather difficult to see.
- 6) Can the authors comment on the repeatability of the shape-memory i.e., how many times can the sample be compressed and the structural colour recovered?
- 7) What sort of load is required to induce the shape change? How much can these samples take?
- 8) In the SI, line 111 – should read laser not lase in: SEM images of grid structures fabricated with different lase power.

Reviewer #2:

Remarks to the Author:

In this work, Zang et al. have developed a Vero Clear based SMPs (shape memory polymers) photoresist with an optically transparent thermosetting polymer resin containing acrylate functional groups. The 4D printing of these SMPs was carried out via utilizing two-photon polymerization lithography. The printed structures are fully characterized by dynamic mechanical analysis (DMA), rheometry, scanning electron microscopy (SEM), ellipsometry and optical microscopy. Finite Difference Time Domain (FDTD) simulation analysis was also carried out to calculate theoretical spectra of the grid structure for direct comparison with the experimental data. Interestingly, a range of structural colours were possible by controlling the geometry of the crosslinked SMP structures. The deformation of the structures at elevated temperatures (80 °C) flattens the nanostructures and makes them colourless. Heating aids in the recovery to the original geometry and colour of nanostructures. Overall, the current study is a submicron demonstration of 4D printing and the ability to change the geometry and optical properties of printing SMPs in response to temperature variation as a function of time is successfully demonstrated. The authors also note that they have produced the smallest feature sizes and highest print resolutions achieved via additive manufacturing of a SMPs, including the first time experimenting with grid structures.

I recommend this manuscript to be accepted after some minor revision. Hopefully, this approach will open many potential applications of 4D printing in the high precision submicron domain and will also overcome the previously reported challenges.

1. The authors use measurements of the mechanical properties of macroscopic film samples in order to choose a suitable resist composition for TPL. They state (SI, page 3, line 47): "The mechanical properties of TPL patterned 47 structures are likely to be similar to that of the bulk film." I would challenge that statement by saying that due to the difference in initiation mechanism (single-photon vs. two-photon), the degree of photoinitiation and thus the amount of crosslinking will be different, leading to a difference in the mechanical properties. The bulk properties can thus not necessarily be extrapolated to microstructures. The authors should include discussion/data addressing this.

2. The authors state (page 5, line 135): "As h_1 only affects the phase of light and does not contribute to the change of colour, it is fixed at $\sim 4 \mu\text{m}$ to raise the grids above the substrate making it easier to compress. The two parameters h_2 and ww_1 can be varied by controlling the write speed, laser power and number of grid layers."

The thickness of the base layer will surely be affected by the writing parameters as much as the other dimensions. Did the authors account for this, or did it not matter since it does not contribute to the change of the colour? If the latter is the case, how can it be said that the thickness was "fixed"?

Reviewer #1 (Remarks to the Author):

The paper by Zhang and co-workers entitled ‘Structural Multi-Colour Invisible Inks with Submicron 4D Printing of Shape Memory 2 Polymers’ describes an exciting new development of nanostructured polymer films that consist of sub-micron features that give rise to a reflected colour. This structure-induced colour can be switched off by subjecting the material to mechanical pressure so as to flatten the film, which can be subsequently recovered by heating the sample to the glass transition temperature. Two photon lithography is used to achieve the submicron features when used with a new polymer resist developed by the authors. Previously, the same team has demonstrated structural colour from high-aspect ratio pillars, but the results presented in this manuscript appear to show a sufficiently novel approach using the grid-like structures.

The study appears to be of a very high quality: the manuscript is very well written, free from jargon, and easily understandable. The figures have also been assembled and prepared with great care. A range of structural colours is demonstrated, which is rather impressive, and there is a good level of detail in terms of the variation in the observed colour with the architecture of the grid and the authors have carried out substantial work to interpret the results. The authors have clearly thought about how the structural colour is influenced by the dimensions of the grid. The experimental data of the structural colour for different nanostructure geometries are also supported, qualitatively by FDTD simulations.

I recommend this manuscript for publication, but suggest that the following minor points are considered first:

1) To remove the colour by applying pressure, the material has to be first heated and then cooled in the presence of an external load, which appears to be somewhat limiting from an applications point of view. Could the authors comment on this? It could be that the key application is that this offers a neat way to validate whether a material has been subjected to high temperatures, which is of importance in the transportation of sensitive goods.

Response: We thank the reviewer for suggesting a potential application of this study. Indeed, we have thought of a similar application. We have added a discussion in the last paragraph of the ‘submicron shape memory polymer’ part in the manuscript as “This demonstration could offer a simple way to monitor whether a material has been subjected to high temperatures, which is of importance in the transportation of sensitive goods.” (Line 289)

Shape memory polymers (SMPs) based 4D printing is widely used due to its wide accessibility of materials and simple way of programming. However, the external load and heat during the programming process may limit the applications in some cases where it is not very convenient or even impossible to apply load and heat. We have performed additional studies using a standard nanofabrication equipment, i.e., nanoimprint lithography, to first “program” the sample. Doing so allows wafer-scale programming of the SMP.

To address the reviewer’s concern, we added the following in the discussion section of the manuscript: “SMPs based 4D printing is widely used due to its wide accessibility of materials and simple way of programming.” (Line 310)

“In some situations where it is not very convenient or even possible to apply load and heat, other methods to induce strain can be considered, e.g., voltage, light, and magnetic field to achieve contactless and precise control of the programming.” (Line 317)

2) The statement on line 114 is rather strong. I am not certain this is the lowest feature size reported, although I cannot immediately point to the literature demonstrating smaller feature sizes. This needs to be checked thoroughly.

Response: We thank the reviewer for this comment. We have done as thorough a check as possible of the literature. While we did not find reports of higher resolutions with SMP, we have added a qualifier to the claim and changed the statement from “ To the best of our knowledge, these are the smallest feature sizes and highest print resolutions achieved via additive manufacturing of a SMP ” to “**This resolution for additive manufacturing of a SMP is an order of magnitude higher than traditional high-resolution printing methods such as DLP^{15,16} and SLA^{17,18}**” (Line 114) in the manuscript.

3) Presumably the polarisation of the writing laser was not of importance in terms of the features that were written?

Response: The laser source in the two-photon lithography system we used (Photonic Professional GT, Nanoscribe GmbH, Germany) is linearly polarized as stated in the official manual. Linear polarization may not be preferred since it could lead to polarization-dependent printed voxel structures, and this polarization-dependent effect has been theoretically and experimentally demonstrated in literature (Appl. Phys. Lett., 2000, 77(5): 612-614; Appl. Phys. Lett., 2003, 83(5): 819-821). For Nanoscribe, the laser beam is conditioned and directed through optics that are tailored for high print performance, thus the 3D printer is configured to minimize influence of polarization. In our experiments, we did not observe a significant polarization-dependent effect when fabricating pillar structures (Nat. Commun., 2019, 10(1): 1-8) and the grid-like structure (this study).

4) Why was the small power range (30-35 mW) chosen?

Response: We have tested a wider range of laser power, as shown in the additional data for the determination of laser power and write speed in Supplementary Information part 3 and new Figure S3. Decreasing the laser power results in some colourless structures in the high write speed regions (Figure S3 I-III), which can result from insufficient crosslinking density and mechanical strength of the grids above the base to survive either during printing or developing. While increasing the laser power can also generate different colours except the ones with low write speeds (0.5-0.7 mm/s) and high laser power (44 mW) due to over exposure (Figure S3 IV-VI), inducing non-uniformity and defects in the colours. We also noted that either increasing or decreasing the laser power would not generate new colours significantly different from the ones we have shown in Figure 2a. Hence, relatively narrow ranges of laser power (30-35 mW) and write speed (0.5-1.4 mm/s) were chosen as “safe” parameters in Figure 2a. In other words, the process window is not particularly large. The observable colour changes within the narrow range of power shows that. The discussions have been highlighted in red in the revised manuscript.

Manuscript, Line 139:

“**See Supplementary Information part 3 on a discussion of laser power and write speed to get different colours.**”

Supplementary Information, Line 95:

“Relatively narrow ranges of laser power (30-35 mW) and write speed (0.5-1.4 mm/s) were chosen for Figure 2a in the manuscript. Decreasing the laser power results in some colourless structures in the higher write speed regions (Figure S3 I-III), which can be resulted from lack of polymerization ratio and mechanical strength of the grids above the base to survive either during printing or developing. While increasing the laser power can also generate different colours except the ones with low write speeds (0.5-0.7 mm/s) and high laser power (44 mW) due to over exposure (Figure S3 IV-VI). We also noted that either increasing or decreasing the laser power would not generate new colours significantly different from the ones we have shown in Figure 2a. Hence, relatively narrow ranges of laser power (30-35 mW) and write speed (0.5-1.4 mm/s) were chosen as “safe” parameters in Figure 2a. In other words, the process window is not particularly large. The observable colour changes within the narrow range of power shows that.”

Figure S3. Optical transmittance micrographs of a printed colour palette for a constant pitch of 2 μm but varying write speed and nominal height h_2 for a wide range of laser power.

5) I found the text in Figures 3b and 3c rather difficult to see.

Response: The colourmaps have been updated in Figure 3b and 3c to increase the contrast between text and figures as shown in the figure below:

Figure 3. Finite Difference Time Domain (FDTD) analysis of the grid structure (a) Measured and FDTD simulated transmittance spectra of structures with different nominal height h_2 (from the black dashed rectangle in Figure 3a ranging from 1.2 μm to 2.7 μm). Marked positions $\lambda_1=490$ nm and $\lambda_2=710$ nm are used for FDTD field analysis in Figure 3b-e; (b)-(c) Cross section view of near field normalized electric field phase and amplitude for a grid structure (laser power: 30 mW, write speed: 1mm/s, nominal grid height: $h_2=2.7$ μm) at dip transmittance 490 nm and peak transmittance 710 nm wavelength respectively; (d)-(e) Top view of far field normalized electric field amplitude for the above grid structure at dip transmittance 490 nm and peak transmittance 710 nm wavelength respectively; the white circle represents collection field for the microscope used in this work (NA=0.2, CA=11.5°); (f) Simulated transmittance spectra for structures with different linewidth w_1 (the colours of the spectrum lines were mapped from the corresponding spectra).

6) Can the authors comment on the repeatability of the shape-memory i.e., how many times can the sample be compressed and the structural colour recovered?

Response: To partly address this question, we performed additional experiments to cycle the SMP up to 10 \times but not to failure, as that would probably require 100s of cycles and is beyond the scope of this paper. The repeatability of the shape memory effect is discussed in Figure 4f in the manuscript. To further investigate how many cycles the sample can stand as the reviewer suggested, we programmed the samples at a fixed pressure and temperature controlled using a Nanonex NX-2006 nanoimprint machine for 10 cycles in Supplementary Information part 12, and the results are given in Figure S11b. No obvious change of colour was observed even after 10 programming cycles. To further increase the service cycles (i.e., more than 1000 cycles), specific chemicals such as monomers with strong π - π interactions and massive chain entanglements (Sci. Rep., 2016, 6: 33610) should be considered as potential candidates for the development of new photoresists in the future studies. The discussion above has been added in line 278 in the revised manuscript.

Manuscript, Line 278:

“Based on this programming process, the repeatability of the shape memory process was further checked, and no obvious change of colour was observed even after 10 programming cycles (Figure S11b). To further increase the number of program-recovery cycles (e.g., beyond

1000 cycles), other specific chemicals such as monomers with strong π - π interactions and massive chain entanglements⁵⁷ could be considered in future studies.”

7) What sort of load is required to induce the shape change? How much can these samples take?

Response: To quantify the influence of stress on the programming process and in an attempt to determine the breaking point, we performed an additional programming process (Supplementary Information part 12) using a nanoimprint machine that enables more control and a wider range of pressure (Figure S11a). As the applied stress increases to 26 psi (~179 kPa), the structure turned colourless (Figure S11b). While a high stress (~1351 kPa) leads to the irreversible collapse around the edges (Figure S11c). Note that in situ micron and nano scale compression experiments (Nat. Nanotechnol., 2019, 14(8): 762-769) at controlled temperatures will need to be conducted to directly determine the relation between stress and strain during deformation of the structure. We added this discussion in Line 270 in the revised manuscript and Supplementary Information part 12.

Manuscript, Line 270:

“To quantify the influence of stress on the programming process and determine the breaking point, we performed an additional programming process (Supplementary Information part 12) using a nanoimprint machine, which enables more control and a wider range of pressures (Figure S11a). As the applied stress increases to 26 psi (~179 kPa), the structure turned colourless (Figure S11b). While a high stress of up to 196 psi (~1351 kPa) leads to the irreversible collapse around the edges (Figure S11c). Note that further in situ micron and nano scale compression experiments⁵⁶ at controlled temperatures will need to be conducted to directly determine the relation between stress and strain during deformation of the structure. ”

Supplementary Information, Line 191:

“To quantify the influence of the stress on the programming process, we performed an additional programming process using a Nanonex NX-2006 machine (Figure S11a). In this set up as shown in Figure S11a, the structures printed on a glass substrate was put between two membranes fixed by the holder. A layer of Teflon film was put between the glass and the upper membrane to avoid contamination during process. After this setup, the holder was put into the chamber where the pressure and temperature can be controlled. Inside the chamber, the space between the membranes was first pumped to vacuum, then the sample was heated up to 50 °C, and then different pressure was applied to the membranes for 30 s. With the pressure maintained, the sample was cooled down to 22 °C (room temperature) to get the deformed configuration. The recovery process was induced by heating the sample to 80 °C by a heat gun. Note that here 50 °C (~10 °C above T_g) was chosen as the elevated temperature to apply stress, which is 30 °C lower than that in the programming process by hand in Supplementary Information part 11. This is because the whole programming process by this Nanonex NX-2006 machine was ~ 5 mins, while the whole programming process by hand was only ~30 s, and we found that it is safer to use a lower programming temperature (50 °C) to avoid irreversible deformation for a relatively long loading and unloading time.

Using this programming process, a structure was programmed under different loads. As the applied stress increases to 26 psi (~179 kPa) or higher, the structure turned to colourless (Figure S11b). While a high stress of up to 196 psi (~1351 kPa) leads to the irreversible collapse around the edges (Figure S11c). Note that in situ micron and nanoscale compression experiments⁵ at controlled temperatures will need to add to directly determine the relation between stress and strain during deformation of the structure.”

Figure S11. Programming process by nanoimprint (a) A photo of the setup of the Nanonex NX-2006 machine; (b) Change of colour as a function of applied pressure; (c) A recovered structure after 196 psi pressure applied in the programming process.

8) In the SI, line 111 – should read laser not lase in: SEM images of grid structures fabricated with different lase power.

Response: Thanks, and the text is corrected as suggested.

Reviewer #2 (Remarks to the Author):

In this work, Zang et al. have developed a Vero Clear based SMPs (shape memory polymers) photoresist with an optically transparent thermosetting polymer resin containing acrylate functional groups. The 4D printing of these SMPs was carried out via utilizing two-photon polymerization lithography. The printed structures are fully characterized by dynamic mechanical analysis (DMA), rheometry, scanning electron microscopy (SEM), ellipsometry and optical microscopy. Finite Difference Time Domain (FDTD) simulation analysis was also carried out to calculate theoretical spectra of the grid structure for direct comparison with the experimental data.

Interestingly, a range of structural colours were possible by controlling the geometry of the crosslinked SMP structures. The deformation of the structures at elevated temperatures (80 oC) flattens the nanostructures and makes them colourless. Heating aids in the recovery to the original geometry and colour of nanostructures. Overall, the current study is a submicron demonstration of 4D printing and the ability to change the geometry and optical properties of printing SMPs in response to temperature variation as a function of time is successfully demonstrated.

The authors also note that they have produced the smallest feature sizes and highest print resolutions achieved via additive manufacturing of a SMPs, including the first time experimenting with grid structures.

I recommend this manuscript to be accepted after some minor revision. Hopefully, this approach will open many potential applications of 4D printing in the high precision submicron domain and will also overcome the previously reported challenges.

General comment

1. The authors use measurements of the mechanical properties of macroscopic film samples in order to choose a suitable resist composition for TPL. They state (SI, page 3, line 47): “The mechanical properties of TPL patterned 47 structures are likely to be similar to that of the bulk film.” I would challenge that statement by saying that due to the difference in initiation mechanism (single-photon vs. two-photon), the degree of photoinitiation and thus the amount of crosslinking will be different, leading to a difference in the mechanical properties. The bulk properties can thus not necessarily be extrapolated to microstructures. The authors should include discussion/data addressing this.

Response: We agree with the reviewer that the mechanical properties for samples cured from TPL and UV light are different. Specifically, the mechanical properties of the TPL and UV patterned structures may be different due to the different degrees of polymerization (Opt. Lett., 2014, 39(10): 3034-3037), alignment of the polymer chains (Nat. Commun., 2018, 9(1): 1-8), and size effects (Sci. Rep., 2015, 5: 17152). Micro and Nanoscale DMA (Appl. Phys. Lett., 2006, 88(13): 131901), compression (Nat. Nanotechnol., 2019, 14(8): 762-769) and tension (J. Mater. Res., 2005, 20(7): 1769-1777) analyses need to be conducted to directly determine the mechanical properties. Nonetheless, the test data provided in our manuscript Supplementary part 2 can still be used as a rough guide to study the trends of mechanical properties as a function of chemical compositions. We used UV light to post cure the printed samples as mentioned in the methods section of the manuscript to minimize the difference of mechanical

properties between UV and TPL fabricated samples. We added discussions on this in both Supplementary Information part 2 and the discussion section of the manuscript.

Manuscript, Discussion section, Line 319:

“Also, micron and submicron scale mechanical tests^{57,60,61} considering the degree of polymerization⁶², alignment of the polymer chains⁶³ and size effects⁶⁴ should be implemented to understand the micromechanical behaviour of the printing.”

Supplementary information, Line 47:

“The mechanical properties of the TPL and UV patterned structures may be different due to the difference of degree of polymerization¹, alignment of the polymer chains² and size effects³. Micro and Nanoscale DMA⁴, compression⁵ and tension⁶ need to be conducted to directly determine the mechanical properties. Nonetheless, the test data provided here can be used as a rough guide to study the trends of mechanical properties as a function of chemical compositions. We used UV light to post cure the printed samples as mentioned in the methods section of the manuscript to minimize the difference of mechanical properties between UV and TPL fabricated samples.”

2. The authors state (page 5, line 135): “As $h1$ only affects the phase of light and does not contribute to the change of colour, it is fixed at $\sim 4 \mu\text{m}$ to raise the grids above the substrate making it easier to compress. The two parameters $h2$ and $ww1$ can be varied by controlling the write speed, laser power and number of grid layers.”

The thickness of the base layer will surely be affected by the writing parameters as much as the other dimensions. Did the authors account for this, or did it not matter since it does not contribute to the change of the colour? If the latter is the case, how can it be said that the thickness was “fixed”?

Response: As the reviewer has correctly pointed out, the thickness of the base layer will definitely be affected by the writing parameters as much as the other dimensions. To clarify, we fixed the exposure conditions for all the base layers in Figure 2-4 and varied only the exposure conditions for the nanoscale mesh structures. The thickness of the base layer was fixed by printing all bases using the same laser power (35 mW), write speed (2mm/s), and number of layers (10 layers), as provided in Table S1. The write speed of the base layer is faster than that of the grids because the pitching distance is much smaller. To address the reviewer’s concern, we have added this information in the revised manuscript (Line 136) as “it is fixed at $\sim 4 \mu\text{m}$ by fixing the laser power (35 mW), write speed (2 mm/s) and number of writing layers (10 layers)” to make it clearer.

Reviewers' Comments:

Reviewer #1:

Remarks to the Author:

The authors have comprehensively responded to all of the comments raised and I recommend that the article be published in Nature Communications. This is a very nice piece of work.

Reviewer #2:

Remarks to the Author:

The authors addressed all points in a clear, thorough and convincing manner. The manuscript as presented is now ready for publication.

Christopher Barner-Kowollik

Reviewer #1 (Remarks to the Author):

The authors have comprehensively responded to all of the comments raised and I recommend that the article be published in Nature Communications. This is a very nice piece of work.

Response: We thank the reviewer for the time and effort to review this work.

Reviewer #2 (Remarks to the Author):

The authors addressed all points in a clear, thorough and convincing manner. The manuscript as presented is now ready for publication.

Christopher Barner-Kowollik

Response: We thank the reviewer for the time and effort to review this work.